# Overview of the Mechanisms of Oxidative Stress: Impact in Inflammation of the Airway Diseases

**DOI:** 10.3390/antiox11112237

**Published:** 2022-11-13

**Authors:** Giusy Daniela Albano, Rosalia Paola Gagliardo, Angela Marina Montalbano, Mirella Profita

**Affiliations:** Giusy Daniela Albano, PhD Istituto di Farmacologia Traslazionale (IFT), Section of Palermo, Italy National Research Council of Italy (CNR), Via Ugo La Malfa 153, 90146 Palermo, Italy

**Keywords:** oxidative stress, natural and synthetic antioxidants, asthma, COPD, lung

## Abstract

Inflammation of the human lung is mediated in response to different stimuli (e.g., physical, radioactive, infective, pro-allergenic or toxic) such as cigarette smoke and environmental pollutants. They often promote an increase in inflammatory activities in the airways that manifest themselves as chronic diseases (e.g., allergic airway diseases, asthma, chronic bronchitis/chronic obstructive pulmonary disease (COPD) or even lung cancer). Increased levels of oxidative stress (OS) reduce the antioxidant defenses, affect the autophagy/mitophagy processes, and the regulatory mechanisms of cell survival, promoting inflammation in the lung. In fact, OS potentiate the inflammatory activities in the lung, favoring the progression of chronic airway diseases. OS increases the production of reactive oxygen species (ROS), including superoxide anions (O_2_^−^), hydroxyl radicals (OH) and hydrogen peroxide (H_2_O_2_), by the transformation of oxygen through enzymatic and non-enzymatic reactions. In this manner, OS reduces endogenous antioxidant defenses in both nucleated and non-nucleated cells. The production of ROS in the lung can derive from both exogenous insults (cigarette smoke or environmental pollution) and endogenous sources such as cell injury and/or activated inflammatory and structural cells. In this review, we describe the most relevant knowledge concerning the functional interrelation between the mechanisms of OS and inflammation in airway diseases.

## 1. Introduction

Asthma and Chronic Obstructive Pulmonary Disease (COPD) are usually distinct; however, these lung diseases have overlapping features. They are inflammatory diseases of the airways, and both asthma and COPD have two common characteristics: inflammation and airway obstruction. The increase in inflammation in asthma and COPD occurs through the activation and alteration of normal activity of both structural cells (epithelial cells, fibroblast, etc.) and infiltrating cells (eosinophils, neutrophils, lymphocytes, etc.). However, the precise mechanisms by which the inflammatory response is regulated in each of these diseases are not clear. Environmental factors such as allergen, cigarette smoke or pollutants can drive the origin of chronic inflammatory lung diseases affecting cellular and molecular mechanisms of oxidative stress (OS). In this scenario, the knowledge of the cellular and molecular mechanisms regulated by OS could be useful to develop new pharmacological treatments useful for the resolution of inflammation in chronic inflammatory diseases of the lung such as severe asthma and COPD.

An imbalance between prooxidative and antioxidative processes define the cell OS in the airway inflammation. The enzymatic and non-enzymatic pathways produce reactive oxidative species (ROS) and compounds related to OS and affect the activities of cells and tissues of the airways. Therefore, the OS is a dynamic and continuous process linked to a wide range of adverse biological effects such as the production of prooxidants, airway infiltration of inflammatory cells, metabolic deregulation and reduced levels of antioxidants. Nonetheless, the lung has several mechanisms to prevent an excessive degree of OS associated with adaptive responses and with resolution of inflammation. The increased levels of OS in the airway promote the progression of disease toward higher disease severity, damaging the lung function and the response to the conventional drug used in the treatment of the disease. However, often the treatment of inflammatory airway diseases with an antioxidant as an additional drug is not sufficient to obtain the welfare of the patients. However, a therapeutic action of the antioxidants might be used in the treatment of asthma and COPD by an approach that considers the individual and environmental risk factors.

## 2. Mechanism of Oxidative Stress

The definition of OS indicates “an imbalance between pro-oxidants and antioxidants with concomitant dysregulation of redox circuits and macromolecular damage” [1]. OS involves the chemistry of the reactions of so-called reactive oxygen species (ROS), reactive nitrogen species (RNS), reactive lipid species and free radicals [2]. ROS and RNS are metabolites of oxygen and nitrogen that have been partially reduced and have a high reactivity and oxidizing capacity. ROS and RNS are composed of free radicals such as superoxide radicals (O_2_^−^), hydroxyl radicals (OH) and nitric oxide (NO) radicals, as well as non-free radicals such as hydrogen peroxide (H_2_O_2_) and peroxynitrite (ONOO) [3]. The free radical is a chemical species with an unpaired electron in its valence shell, highly unstable and reactive, that interacts with and damages the cellular biomolecules (proteins, lipids, DNA and carbohydrates) [4]. NO is a lipophilic gaseous transmitter also secreted by the epithelial cells [5].

ROS are produced by normal cells at low concentrations to ensure cellular signaling and prolonged homeostasis [6]. ROS and RNS can have a dual function, depending on whether they are present at low or high concentrations. In the first case, they are involved in redox cell signaling by regulating some normal physiological processes such as phagocytosis, vasodilation, tissue repair and regeneration [4,7,8]. On the other hand, at high concentrations, they lead to the onset of cell and tissue damage that can cause the onset of inflammatory diseases (such as respiratory diseases and cancer) [9,10,11,12,13]. There are two sources of ROS and RNS in the lungs: an endogenous source represented by cells that produces free radicals and an exogenous source whose main representatives are atmospheric particulate matter and cigarette smoke [14]. In general, ROS/RNS are produced as intermediates or by products of cellular metabolism catalyzed by enzymes localized in different organelles, primarily plasma membrane, cytosol, mitochondria, peroxisomes and endoplasmic reticulum [15].

The key enzymes that produce ROS/RNS in the lungs include NADPH oxidase, myeloperoxidase (MPO), xanthine oxidase and NO synthase (NOS). NADPH oxidases (NOX) are a class of a multicomponent transmembrane enzyme complex that transports electrons across biological membranes to reduce oxygen to superoxide [8,10,11,12,16]. There are seven isoforms of NOX—(NOX-1, NOX-2, NOX-3, NOX-4, NOX-5, Dual oxidases (DUOX)-1 and -2) [17]. Many of them are expressed in various cell types of human lung [18,19]. NOX-2 is localized in the phagosome of macrophages and neutrophils, the first cells of the immune systems recruited at the site of infections producing ROS to kill the phagocytized bacteria [20].

Myeloperoxidase (MPO) belongs to the heme peroxidase family and is abundantly expressed in neutrophils and macrophages. It generates hypochlorous acid (HOCl), a damaging reactive oxygen species (ROS) utilizing hydrogen peroxide (H_2_O_2_) and chloride (Cl^−^) and, therefore, it is an important enzyme in the host defense against bacteria, viruses and fungi [21]. The production of HOCl and other reactive oxidants by MPO may also damage tissues. The formation of chlorotyrosine residues that MPO induced is used as a marker of neutrophilic inflammation [22]. In addition, MPO uses as a substrate NO and nitrite, leading to the formation of nitrogen dioxide (NO_2_) that induces a post-translational modification contributing to inflammatory disease through protein nitration, lipid peroxidation and the oxidation of tyrosine yielding nitrotyrosine [23].

Other enzymes that catalyze chemical reactions that are ROS-generating are xanthine oxidase (XO), localized in the plasma membrane, and cytosol, which under normoxia catalyzes oxidation of hypoxanthine to xanthine and to uric acid; however, under hypoxic conditions, hypoxanthine is formed from adenosine triphosphate, and oxygen is reduced to hydrogen peroxide and superoxide anion [24,25].

The lipoxygenases (LOXs) are non-heme iron enzymes catalyzing dioxygenation of polyenoic fatty acids yielding hydroperoxyl derivatives including hydroperoxyeicosatetraenoic acids (HPETEs); cyclooxygenases (COXs) are bifunctional enzymes (having both COX and peroxidase activities) that release arachidonic acid (AA) from membrane phospholipids and catalyze conversion of AA to prostanoids; and nitric oxide synthase (NOS) is a heme-containing monooxygenase that catalyzes oxidation of L-arginine to citrulline and nitric oxide (NO) [26]. Moreover, NO is involved in multiple biological processes including vasodilation and inflammatory processes, and in several airways’ epithelial physiologic functions as mucociliary function and ciliary frequency, in the modulation of inflammation by inflammatory mediators release and bronchial epithelial barrier integrity [27]. There are three isoforms of NOS, namely, NOS1 (neuronal NOS, nNOS), NOS2 (endothelial NOS, eNOS) and NOS3 (inducible NOS, iNOS). Pulmonary cells constitutively express NOS1 (in neurons and endothelial cells) and NOS2 (in the human airway epithelium, lung endothelium and alveolar macrophages), while NOS3 (in bronchiolar epithelial cells and the endothelium) is elevated in the lung during inflammation [28,29]. NOS catalyzes the transformation of l-arginine, molecular oxygen (O_2_) and NADPH-derived electrons to nitric oxide (NO) and l-citrulline. Under some conditions, however, NOS catalyzes the reduction of O_2_ to superoxide (O_2_^−^) instead, a phenomenon that is generally referred to as uncoupling [30].

## 3. ROS Inflammation and Autophagy/Mitophagy Processes

ROS have impacts on several signaling pathways and mechanisms. In response to growth factors and cytokines, they affect a variety of cell processes [31] associated with the development of several pathogenetic mechanisms, including inflammatory processes and tumorigenesis. ROS generate DNA damage, immune response, immune evasion, signaling pathway regulation involved in the control of autophagy, and apoptosis, angiogenesis and drug resistance [32]. The massive production of ROS leads to inflammation status capable of altering the tumor microenvironment (TME), promoting DNA damage as well as the upregulation of growth factors, cytokines and genes involved in cell survival, underlining their impact on several signaling pathways and mechanisms [33,34]. In response to growth factors and cytokines, ROS can act as secondary messengers for specific signaling pathways as well as regulatory molecules for gene expression [32]. In most of the lung disorders, the primary source of ROS excess are the mitochondria [5,10,35]. Superoxide produced by mitochondria is a result of incomplete reduction of oxygen to water due to leakage of electrons by a mitochondrial respiratory chain, which is represented by four complexes composed of proteins, labeled I through IV: respiratory complexes I (aka NADH:ubiquinone oxidoreductase), II (aka succinate-coenzyme Q reductase), III (aka ubiquinol:cytochrome c oxidoreductase) and complex IV (aka cytochrome c oxidase) [5,35,36]. In the mitochondrial electron transport chain, the electrons are being passed from Complex I or Complex II to Complex III via ubiquinone, and some of these electrons can escape and react with oxygen to form superoxide. The superoxide can convert to hydrogen peroxide through enzymes’ superoxide dismutase, which can then exit the mitochondria. In the cytoplasm, metabolism reactions such as those of the cytochrome P450 family of enzymes (CYP) produce ROS [37,38].

Mitochondria are, also, essential organelles within the cells where most ATP is produced via oxidative phosphorylation (OXPHOS) [39,40]. Mitochondrial ROS (MtROS) stimulates redox-sensitive transcription factors such as hypoxia-inducible factor-1 (HIF-1), NF-kB and pro-inflammatory cytokines to activate inflammatory caspases (caspase-1 and -12) and to promote tumor progression activating TNF-converting enzymes, which is required for the progression of inflammation [41,42,43,44]. As mentioned, the principal sites of ROS production are mitochondria, the principal site of the regulation of autophagy mechanisms. Autophagy plays a major role in the first line of defense against antioxidant damage and regulates cell homoeostasis via the lysosomal pathway by eliminating and recycling proteins and organelles within the cells [45]. ROS influences autophagy both directly and indirectly. Direct regulation occurs when key proteins Atg4, Atg5, and Beclin are damaged in their autophagic function. Instead, indirect ROS regulation involves alteration of signaling pathways that can induce autophagy such as the JNK, p38. [41,42]. Moreover, the ROS range of actions encompasses the direct activation of the redox sensor nuclear factor erythroid 2-related factor (Nrf2). Nrf2 is a primary cell survival regulator involved in the control of antioxidant mechanisms, drug metabolism and anti-inflammatory detoxification, and promotes cancer progression as well as protecting the cells from OS and DNA damage [46]. In fact, Nrf2 serves two purposes: it promotes inflammation and cancer by enhancing OS and mitochondrial dysfunction, and it strengthens the antioxidant system to suppress tumors and inflammation [47].

OS is reduced through the Nrf2 / Keap1 (Kelch Like ECH Associated Protein 1) and SQSTMI / p62 pathways that induce the expression of antioxidants regulated by Nrf2. [48]. The p62 pathway increases binding affinity to Keap1 and induces selective autophagy activating Nrf2 that consequently activates the antioxidant enzymes such as catalase, glutathione peroxidase, peroxire-doxin and glutathione [49]. OS is, also, regulated by carbonyl reductase 1 (CBR1), an enzyme that regulates Nrf2 expression. However, Nrf2 hyperactivation may create an environment favoring both normal and malignant cells, protecting them from OS [50]. Moreover, in the presence of elevated ROS, the deletion of some autophagy genes such as ATG5, ATG7 or BECLIN1 establishes defective autophagy or autophagy inhibition, which promotes tumor initiation and progression through chronic OS, inflammation and tissue damage. [51].

High levels of ROS impair the mitochondrial function, and consequently they trigger a signal of self-removal through a process called mitophagy.

The mitophagy process acts to remove dysfunctional mitochondria by fusion with lysosomes and to control the number of mitochondria maintaining energy metabolism stability [51,52,53]. The activation of phosphatase and tensin homology deleted on chromosome 10 (PTEN) regulated through the putative kinase 1 (PINK1)/Parkin pathway can induce the ROS to trigger the mitophagy mechanism [54]. Under normal physiological conditions, cellular prion protein (PrPc) binds PINK1 through ubiquitin kinase and enters the mitochondrial inner membrane, degrading it. Under oxidative stress, PINK1(PTEN-induced kinase 1) can act as a molecular sensor of damaged mitochondria; PINK1 facilitates aggregation and clearance of depolarized mitochondria through interactions with Parkin and possibly Beclin1 that are involved in the clearance of damaged mitochondria [53,55]. Ubiquitous mitochondria are encapsulated to form mitophagosome, which are fused with lysosomes and reduced by hydrolases [53,56] (Figure 1).

## 4. Oxidative Stress in Asthma

Different immune endotypes associated with eosinophilia (atopic or non-atopic Th2 type immune response) or neutrophilia (non-Th2 response) characterize the heterogeneity of inflammation in asthma. Inflammatory processes in asthma accompany airway hyperresponsiveness and guide airway remodeling. Furthermore, OS is present in the airways and in the blood circulation of asthmatic patients; however, it is not clear whether OS is the cause or the consequence of chronic changes in inflammation and in the related remodeling. Moreover, the oxidant “burst” in asthma is probably a self-propagating nonspecific process due to the concomitant action of several inflammatory pathways. Asthma mediators such as cytokines, lipid mediators, adhesion molecules and granulocyte granule proteins are potential stimuli or promoters of ROS. In addition to endogenous mechanisms, environmental factors such as allergens and air pollutants may induce the increase of ROS production in the airways.

The increase in oxidative damage of biomolecules and the presence of high concentrations of arachidonic acid oxidation products such as 8-isoprostane in exhaled breath condensates (EBC), as well as the increased levels of lipid peroxidation compounds in EBC of asthmatic patients, support the presence of high levels of OS in asthma [57,58], often associated with diseases’ severity, and amplify the inflammatory response reducing responsiveness to corticosteroids.

The imbalance between the production of pro-oxidants (e.g., ROS or RNS) and antioxidant defense mechanisms generates OS in the body. As discussed above, the endogenous sources of ROS are mitochondria, peroxisomes, endoplasmic reticulum of the immune (phagocytes, activated eosinophils and neutrophils, monocytes and macrophages) and structural cells (epithelial cells, smooth muscle cells, endothelial cells) or enzymes and enzymatic complexes (e.g., NO synthase, NADPH oxidases, xanthine oxidase). At the same time, ROS can be generated by exogenous sources, such as cigarette smoke, pollutants, allergens, ozone, organic solvents, metals, ultraviolet light, ionizing radiation and some drugs. Both endogenous and exogenous sources of ROS are involved in initiation and activation of intracellular signaling promoting inflammatory and immunological mechanisms in the airways [59].

ROS are strongly reactive and at low concentrations in the tissues. ROS show a short half-life, and this characteristic makes difficult its direct measurement in vivo. The biomarkers of OS in asthma are lipid peroxidation (including malondialdehyde, 8-isoprostane), protein oxidation (protein carbonylation) and DNA damage or general antioxidant capacity. Accordingly, non-invasive biomarkers of OS, such as ROS, markers of lipid peroxidation, NO metabolites and organic compounds, are detected in EBC recovered from the airways of asthmatic patients [60,61,62,63,64]. EBC higher levels of H_2_O_2_ and NO indicate increased levels of nitrites (NO_2_) and nitrates (NO_3_) in patients with asthma. Furthermore, the levels of H_2_O_2_ and NO positively correlate with eosinophil counts from induced sputum and inversely correlate with lung function in asthmatic patients [61,65,66]. Finally, EBC levels of 8-isoprostane and malondialdehyde are significantly higher in mild-moderate and severe asthmatics in comparison with healthy subjects, in both adult and paediatric patients [57,67,68,69].

OS is evident at systemic level in asthma. Blood leukocytes from asthmatic patients produce higher O_2_^−^ levels and contain lower GSH levels. High levels of lipid peroxides and nitrites/nitrates, and low GSH concentration and glutathione peroxidase activity are present in patients with asthma [70,71,72]. Moreover, OS markers increase in asthmatics and correlate with disease severity, exacerbation frequency and phenotype [73,74]. Numerous in vitro and in vivo studies indicate ROS as important mediators of asthma pathogenesis. In fact, ROS drive many aspects of diseases as inflammatory responses, airway remodelling and airway hyperresponsiveness [67,75,76,77]. Both eosinophils and neutrophils, via the oxidative burst dependent on O_2_^−^ produced by NOX-2, play a relevant role in ROS production in asthma [77]. In particular, eosinophil peroxidase (EPO) and neutrophilic myeloperoxidase (MPO) convert O_2_^−^ to H_2_O_2_ to induce protein nitration and halogenation [73]. The imbalance of redox activity is observed in airway epithelial cells from asthmatics. DUOX-1 and DUOX-2, and NOX-4 overexpression is observed in airway epithelial cells and airway smooth muscle cells in asthma [78,79]. These enzymes mediate ROS production in the lung.

Oxidation and nitration of manganese-superoxide dismutase (MnSOD) are present in the asthmatic airways, and these modifications correlate with clinical features of asthma severity. Tyrosine nitration and chlorination from eosinophil and neutrophil activation cause inhibition of MnSOD activity and catalase, impairing antioxidant defenses in asthma [72,80,81,82,83,84]. Redox imbalance and redox-dependent mechanisms engage inflammatory mediator release, epithelial damage, reduced lung function and airway hyperresponsiveness in asthma pathogenesis [85,86]. The activation of redox-sensitive pathways such as nuclear factor (NF)-κB, activating protein (AP)-1, phosphatidylinositol 3-kinase (PI3K)/Akt, Janus Kinase/Signal Transducer Activator of Transcription (JAK/STAT) and MAPKs drive transduction and signaling mechanisms involved in asthma [67].

There is evidence that allergens promote ROS production in the lungs indirectly, via immune and structural cells’ activation or by directly intrinsic mechanisms. Pollens are complex structures that in addition to having antigenic properties contain NOX enzymes involved in airway ROS production [75]. DUOX enzymes affect crucial PAMPs and DAMPs activation in airway epithelium of allergic subjects. IL-4 and IL-13 activate non-canonical autophagy and generate DUOX-2 trafficking through apical membrane of airway epithelial cells of the airways [87]. DUOX-2 interacts with TLR2 and TLR4 as well as conversely the activation of TLR2, TLR3, TLR5, and TLR6 drives increased expression of DUOX-1, 2 in airway epithelial cells [88,89]. NOX-1 and NOX-2 are involved in the activation of innate immune effector promoting the immunological cell response in asthma [90]. TLR4 activation upregulates NOX-1 expression via leukotriene B4 receptor-2-dependent mechanisms and ROS-dependent NF-κB activation and Th2 cytokine release [59,91,92].

ROS production in response to allergens increases the expression of the NOX-1 and NLRP3 inflammasome complex, consisting of the NLRP3 receptor, caspase-1 and the adaptor ASC. In addition, it detects exogenous PAMPs and DAMPs and modulates the release of active IL-1β and IL-18 [93]. Moreover, ragweed pollen extract enhances the IL-1β production endotoxin-dependent in human macrophages and dendritic cells via the NLRP3 inflammasome activation NOX-dependent. In this context, the priming of NLRP3 inflammasome complex is also mediated by ROS production [94].

ROS are involved in the orchestration of adaptive immune responses. T-cell maturation occurs via the T cell receptor (TCR)-MHC-antigen complex, and the interaction between co-stimulatory molecules and the antigen presenting cells provides signals to the innate immune cells for the production of cytokines and ROS [95]. Intracellular ROS production due to NOX-2 activation in T cells TCR-mediated regulates T cell lineage commitment and activation [96,97]. Indeed, NOX-2-deficient mice exposed to an allergen have Th2 differentiation of CD4 T cells, lung eosinophilia, goblet cell metaplasia and hyperresponsiveness [98].

The complexity of redox signaling, involved in allergic immune responses and asthma disease, suggests the necessity to identify the phenotype of patients that benefit from the use of pharmacological treatment with an antioxidant drug for an individual approach of precision medicine.

## 5. Oxidative Stress in COPD

COPD is a chronic inflammatory disease associated with an irreversible airway obstruction. COPD represents one of the most important causes of mobility problems and mortality. In fact, it is the third leading cause of death worldwide, with enormous human and social costs. The inflammatory processes of COPD patients are activated and amplified not only via the “conventional” pro-inflammatory pathways, including innate response, but also via the coagulation pathways and via the release of neurotransmitters. Chronic inflammatory processes may determine tissue destruction and interfere with physiological repair mechanisms, promoting tissue architectural alterations, leading in turn to respiratory failure [99,100]. In the Western world, the most important etiologic factor in causing COPD is cigarette smoking with inhalation of combustion products. The sources of OS in the lung also involve gases and ultrafine particulate material, and nanoparticles’ inhalation from industrial and car pollution [101]. The lung is particularly exposed to injury due to environmental factors; however, it is also subjected to endogenous OS generated by mitochondrial respiration, and by the immunological and inflammatory responses to bacterial and viral infections within airways [102]. A range of 15%–20% of smokers develop COPD, and cessation of smoking does not stop disease progression with continued evidence of inflammatory cell recruitment to the lungs and the presence of OS [103].

COPD has as a feature neutrophilia [104]. The neutrophils to sites of injury or infection are activated and induce the release a cytotoxic and proteolytic cocktail that allows effective killing of invading micro-organisms by enhancing ROS and NO generation and subsequently undergoing apoptosis [105]. The excessive recruitment of neutrophils and poor clearance of apoptotic neutrophils by macrophages can cause secondary necrosis, whereby they release lysosomal constituents affecting resident lung cells [105]. This indicates self-perpetuating endogenous inflammation processes in susceptible individuals [106,107] with persistent release of inflammatory mediators such as leukotriene B4 (LTB4) and interleukin (IL)-8 [108], and continued recruitment and activation of neutrophils to the lungs. The release of proteases, free radicals, cytokines and pro-inflammatory mediators from these activated cells plays a key role in COPD airways leading to the destruction of surrounding tissues, a loss of lung elasticity and mucus hypersecretion and emphysema [109,110]. As described above, OS occurs when exposure to free radicals, such as ROS, arising during mitochondrial respiration, cell signaling and in response to tissue injury and pathogens, overwhelms antioxidant defenses. Potential targets for damage by ROS include DNA, proteins and lipids. There is significant theoretical and experimental support for the potential relationship between ROS lung damage and pathogenesis of COPD. The assessment of OS presence in the lungs of COPD patients has been evaluated with a variety of methods, and there is clear evidence of an increase of oxidative burden in COPD compared with non-smoking healthy controls [111].

EBC is a helpful method used to identify OS products found in the airways [112]. Several studies have shown that H_2_O_2_ is significantly increases in the EBC of COPD patients compared to healthy control subjects [113,114], and levels of H_2_O_2_ increase even further during disease exacerbations. Isoprostanes, formed from in vivo free radicals’ peroxidation of arachidonic acid [115], can be measured in EBC and have been found to be higher in COPD [116]. Malondialdehyde (MDA), another product of fatty acid peroxidation, significantly increases in the EBC of COPD patients in comparison to healthy control subjects, asthmatics and subjects with bronchiectasis. Moreover, MDA levels inversely correlate with FEV_1_ [117], suggesting a relationship with disease severity. In addition, in a further study, serum levels of MDA correlated with COPD severity [118].

Using the immunohistochemistry technique, it is possible to identify some products of OS within distinct cellular components of the lungs of COPD patients [119,120]. The cellular key sources of ROS in COPD are represented by neutrophils, monocytes/macrophages and lung epithelial cells. Neutrophils are key effector cells in COPD. They are significantly increased in COPD lungs [121], correlate with disease severity, and their products have been shown to cause several immunopathological and functional features of disease [122,123]. It has been shown that neutrophils from COPD release increased amounts of ROS spontaneously and following stimulation [124,125,126].

Lung epithelial cells of COPD patients are primarily subjected to exogenous OS but also produce endogenous OS products derived from mitochondrial respiration [127]. The NADPH oxidases (NOX) of membrane, the xanthine/xanthine oxidase system and neutrophils derived myeloperoxidase (MPO) are enzymes generating ROS in cellular cytoplasm [111]. NOX produce superoxide anions in the cells. They are weak oxidizing agents that in the presence of NO are rapidly converted in damaging ROS species as H2O2 and hydroxyl radical or the peroxynitrite radical [86]. MPO levels, obtained from activated neutrophils, increase in the lungs of COPD patients. They cause the production of hypochlorous acid, chlorinates protein tyrosine residues contributing to the formation of 3-chlorotyrosine in sputum of COPD patients [128]. Interestingly, intracellular antioxidant defenses can eliminate the damaging ROS species in the airways of healthy subjects, limiting the pro-oxidative effects. Conversely, the mechanisms of antioxidant defenses are overwhelmed in COPD [111].

ROS generation affects the reactive carbonyl production by lipid peroxidation and sugar glycoxidation. The result is aldehydes and protein carbonylation [129]. The “carbonyl stress”, due to the formation of reactive carbonyls and subsequently of protein carbonylation, is associated with the age and chronic diseases in the airway, leading to cell and tissue mechanism dysfunctions. It is present in both smokers and COPD patients and is correlated with disease severity [130]. High levels of XO are detected in COPD patients and correlate well with the levels of cytokine expression in the bronchial mucosal lining fluid of COPD subjects [86,128].

Increased OS drives several mechanisms of COPD pathophysiology with different effects in the lungs. The inflammatory cascades are sensitive to OS. They involve redox-sensitive molecular targets as signaling molecules such as Ras/Rac, Jun-N-terminal kinase (JNK), p38 mitogen-activated protein kinase (MAPK), the transcription factor nuclear factor-κB (NF-κB) and protein tyrosine phosphatases [111]. The treatment with antioxidant GSH (L-Glutathione reduced) reduced IkB Kinase/NF-kB pathway activation in an in vitro study on peripheral blood cells from COPD patients [131].

OS activates the TGF-β signaling pathway and promotes small airway fibrosis [132]. This is the result of the reduction of endogenous antioxidant activities in the lung via the inhibition of Nrf2 signals [133]. OS increases the expression of MMP9, an enzyme primarily involved in elastin degradation and collagen hydrolyzation, and lung emphysema enhancing elastolysis through oxidative inactivation of α1-antitrypsin and the secretory leukoprotease inhibitor [134].

Corticosteroids control gene expression of pro-inflammatory markers impaired in COPD patients, as the result of OS inhibition, via the reduction of histone deacetylase-2 (HDAC2) activity involved in inflammatory gene suppression [135]. HDAC2 function is reduced by OS activation of phosphoinositide-3-kinase (PI3K)-δ, which leads to phosphorylation and ubiquitination of HDAC2 [136] and the formation of peroxynitrite [137]. This results in amplified inflammation and corticosteroid resistance. Currently, there are no clinically available treatments that prevent COPD progression. As discussed, OS is a major driving mechanism for the chronic inflammation, disease progression and exacerbations of COPD, and induces corticosteroid resistance [131,138], so targeting OS is crucial in order to develop important therapeutic strategy in the future [139,140]. Based on these concepts, it is necessary to improve our knowledge of the prevention and on the diagnostic and therapeutic strategies associated with OS in the early steps of the disease.

## 6. Environmental Pollution and OS in the Airways

Environmental pollution derives from combustion processes of human activities (industrial emissions, domestic heating, vehicular traffic) and of natural phenomena (e.g., volcanic eruption) [141,142,143]. The particles present in the air as pollutants are a range of liquid or solid particles present in the atmosphere with different chemical and physical characteristics. The source, the size, the mechanisms of production and the chemical nature generate the different properties of pollutants. Gases such as O_3_, sulfur dioxide (SO_2_), NO, NO_2_, carbon monoxide (CO), carbon dioxide (CO_2_) or volatile organic compounds such as benzene, particulate matter (PM), metals, nitrates, sulphates, organic carbon, microbial components and pollen are present in the air [144,145,146,147]. The size and the chemical composition of pollutants affect the effect environmental pollution on human health [148].

Air pollution generates both acute and long-term effects to human health worldwide [149]. Adult and children subjects absorb air contaminants through the respiratory tract and skin that subsequently reach the bloodstream to target the human organs, causing damage to human health [141,142,143]. Air pollution is associated with a higher risk of premature death due to cardiovascular diseases, stroke, dementia, diabetes, asthma, COPD, lower respiratory tract infections and lung cancer [150,151,152,153,154,155].

The World Health Organization (WHO) reports that PMs, SO_2_, O_3_ and NO_2_ levels are higher than those of the national standard and criterion concentration of many cities. High concentration of particles and toxic compounds dramatically impair the pulmonary functions of human beings, consequently resulting in respiratory disorders or even death [156,157,158]. In fact, long-term exposure to air pollutants affects airway diseases in adults, children and adolescents [159]. The environmental contamination weakens the immune response of the human body and interferes with the defense mechanisms of the lung [160]. Finally, air pollutants carried by the floating aerosol particles and delivered deep in the lung cause fiber hyperplasia of the alveolar wall, promoting lung fibrosis and emphysema involved in COPD processes [161].

The multifactorial respiratory diseases have various risk factors, including OS of the cells related to air contamination [159,162,163] (Figure 2). Air pollution increases inflammation-related cascade and oxidation stress in the lung, vascular and heart tissue. Initially, OS mechanisms are increased in the lung as a protective mechanism involved in the removal of the injurious stimuli by ROS production to activate the cell killing. Furthermore, during the early phase of inflammation, OS mechanisms induce the transcription of anti-oxidant gene stress without cell damage [164]. PM10 affects the production of inflammatory mediators and OS markers in lung epithelial cells and alveolar macrophages. The cytokines and chemokines released in the lung reach the marrow and stimulate the migration of neutrophil precursors into blood circulation [165].

PM, NO_2_ and SO_2_ are potent oxidants, either through direct effects on lipids and proteins or indirectly by the activation of intracellular oxidant pathways [163,166,167,168], and by the production of highly reactive hydroxyl radicals they initiate oxidative DNA damage [169,170]. PM10, PM2.5 and SO_2_ induce short- and long-term effects on lung function [171] and through a variety of mechanisms such as direct oxidative injury promote inflammation in the airways [172]. In this manner, they exert a negative effect in the respiratory system, especially in the cardiopulmonary system [173].

The exposure of a human lung epithelial cell line to PM10 promotes ROS generation and decreases the activity of the antioxidant enzymes (superoxide dismutase and glutathione reductase) [174]. The exposure of primary cultures of human bronchial epithelial cells to PAH (polycyclic aromatic hydrocarbon) adsorbed on PM2.5 induce the persistence of a prolonged inflammation state and affect the OS mechanisms delaying repair processes in injured tissues [175]. PM (ranged from 0.25 to 2.5 μm) and O_3_ are positively associated with exhaled NO. PM 0.25, CO and NO are positively associated with IL-6 and ROS in elderly subjects enrolled [176]. Both Vanadium pentoxide (V2O5), a component of PM derived from fuel combustion (source of occupation-related exposure in humans), and chromium (VI) (detectable in PM2.5) affect oxidative DNA damage in human lymphocytes [177].

Particles bound to benzo(a)pyrene are bioavailable and induce oxidative DNA damage in target organs including the lung [178,179]. Moreover, O_3_ and NO_2_ are usually present in air and promote oxidative stress and DNA damage in the lung tissue [159,163]. Organic compounds adsorbed on particle surfaces promote inflammation through CYP1A1-mediated ROS generation, and favor the release of cytokines after activation of transduction pathways involving MAPK and the transcription factor NFkappaB in human airway epithelial cells [180]. NF-*κ*B, activator protein 1 (AP-1) and CAATT/enhancer binding protein (C/EBP) regulate ROS production and proinflammatory gene transcription (TNF-*α* and IL-8, etc.) in human alveolar and bronchial epithelial cells in response to PM exposure [181,182,183].

Epidemiological data show that patients with asthma and COPD are sensitive to O_3_ exposure and exhibit increased morbidity with a higher risk of mortality [184,185]. O_3_ is a toxic photochemical air pollutant causing respiratory and cardiovascular exacerbation. It increases the cases of mortality and hospital admission rates [186]. O_3_ is a compound that is highly reactive, with the capacity to oxidize proteins and lipids in the lining fluid compartment of the lung [184,185]. O_3_ reacts with biological tissues and causes detrimental health effects such as harming lung function, irritation of the respiratory system and impairing tissues [187]. The oxidized species arising from the reaction between O_3_ contributes to the acute bronchoconstrictor response and hyperresponsiveness in asthmatic subjects [188,189]. Furthermore, O_3_ induces apoptosis, DNA damage and cytotoxicity in human alveolar epithelial type I-like cells and in mice exposed to O_3_ for 6 weeks [190,191]. Finally, ozone induces the OS in the lung, promoting the mechanism of chronic inflammation and emphysema in COPD patients [192].

The environmental exposure to the heavy metals cadmium (Cd), lead (Pb), mercury (Hg), nickel (Ni), metalloid arsenic (As), and transition metal chromium (Cr), induce reactive oxygen species (ROS) production, and OS mechanisms’ activation in various organs [193]. These elements are known to damage human tissues and organs at low concentrations, including the lung [193]. In fact, the heavy metals exhibit their harmful effects, including inflammation and carcinogenicity, and affect the oxidative response in airway epithelial cells [194] and organs in various diseases by Nrf2 signaling [195]. The activation of OS mechanisms in the lung by the exposure of the subjects to heavy metals generates free radicals, reduces antioxidant levels, alters DNA structure and miRNA expression; inhibits ion channels, ATP-ases and other transporters; increases cytoskeleton and cell polarity; or even determines the impairment of endocytosis and intracellular vesicle recycling in airway diseases regulating the mechanisms of cell communication [196,197,198].

## 7. New Pharmacological Perspectives in Airway Diseases: Antioxidants

The antioxidants have different chemical structures that distinguish them in respect to solubility in water (hydrophilic) or fat (hydrophobic or liposoluble). Generally, the hydrophilic antioxidants found in the cytosol or cytoplasmic matrix react with ROS within cells or body fluids (blood serum, extracellular fluid, seminal plasma), while the liposoluble antioxidants are present in cell membranes and are more prone to protect cell membranes from ROS-mediated lipid peroxidation [199,200]. They are defined as “free radical scavengers”, acting as a hydrogen donor, electron donor, peroxide decomposer, singlet oxygen quencher, enzyme inhibitor, synergist, and metal-chelating agent.

The human natural antioxidant defense system is distinguished into two categories: exogenous and endogenous (enzymatic and non-enzymatic) antioxidants. The first class is obtained from diet and the repair of free radical damage from the inside by stimulating cell regeneration; the second class is made by our body and repairs free radical damage from the inside by initiating cell regeneration [201]. Endogenous antioxidants are known as enzymatic and non-enzymatic depending on their activity. Endogenous enzymatic antioxidants consist of glutathione peroxidase, superoxide dismutase, thioredoxin and catalase, while endogenous non-enzymatic molecules include melatonin, bilirubin, uric acid, polyamines and glutathione (GSH) [202]. On the other hand, the group of exogenous non-enzymatic antioxidants includes vitamins E, A and C, flavonoids, carotenoids, plant polyphenols, theaflavin, allyl sulfides, selenium and curcumin [203,204]. Enzymatic antioxidants convert oxidized metabolic products in hydrogen peroxide (H_2_O_2_) and then in water by cofactors as iron, zinc, copper and manganese. Instead, non-enzymatic antioxidants work by blocking free radical chain reactions [205,206,207].

Many studies show that OS is quite prevalent in many diseases, among which is chronic inflammatory lung disease, but also in the systemic circulation of asthmatics due to proximity of pulmonary vasculature with blood capillary network [208]. The lung contains high levels of antioxidant resources to prevent oxidant-induced injury, including both enzymatic and non-enzymatic systems [209]. In the past years, the interest in the use of antioxidants as pharmacological treatment of airway diseases has grown [210].

To foster a close antioxidant network, non-enzymatic and enzymatic antioxidants are needed to be considered at the same time. In the airways, the first line of defense is respiratory-tract lining fluid (RTLF), a thin fluid layer covering the respiratory epithelium [209]. RTLF is constituted by vitamin C, urate, reduced glutathione (GSH), vitamin E, bilirubin, extracellular superoxide dis-mutase (SOD), catalase and extracellular glutathione peroxidase (GPx), and protects from oxidative injury caused by inhaled environmental/endogenous oxidants [209]. Additional antioxidants include mucopolypeptide glycoproteins, caeruloplasmin and Fe-binding proteins [202,211]. Recently, other enzymes playing a crucial role in the antioxidant mechanisms of defense were found in the lung: heme oxygenase-1, small molecular weight redox proteins such as thioredoxins, peroxiredoxins and glutaredoxins [202]. It is known that the pathogenesis of many lung diseases is induced by oxidative imbalance and the generation of ROS [212]. ROS and oxidative imbalance are used as therapeutic targets for therapeutic strategies in the lung [213].

Based on the mechanism of action, we can distinguish three groups of antioxidants’ drugs: (1) enhancers of endogenous antioxidant enzymes, such as superoxide dismutase (SOD), catalase (CAT) and glutathione peroxidase (GPx), which accelerates the conversion and inactivation of free radicals; (2) non-enzymatic scavengers of excess free radicals and lipid peroxyl radicals, which keep the cell membrane intact; and (3) drugs with other mechanisms [214]. N-acetyl cysteine (NAC) is an enzymatic antioxidant drug that acts as (1) a reductant of disulfide bonds, (2) a scavenger of reactive oxygen species and/or (3) a precursor of glutathione biosynthesis. This is a pleiotropic drug with various pharmacologic characteristics [215]. NAC provides cysteine for the increased intracellular production of glutathione. NAC is used as a mucolytic agent since it hydrolyzes the disulfide bonds of mucus proteins and decreases its viscosity, thereby facilitating its clearance and lung secretions [216]. NAC also directly inactivates reactive electrolytes and free radicals in a non-enzymatic manner and maintains the oxidant/antioxidant balance in cells. Moreover, NAC reduces the formation of some proinflammatory cytokines, such as IL-9 and TNF-α [217]. For many years, it was believed that NAC had only mucolytic properties [218], but in more recent times it was discovered that this antioxidant has positive effects as an anti-inflammatory and reduces the acute exacerbations in COPD with greater effect in smokers and in subjects not treated with inhaled corticosteroids [219,220,221]. It is now known that the sum of all these effects describes the effectiveness of NAC.

As we have previously described, NOX is a major source of ROS especially in COPD patients. Several NOX inhibitor drugs have been developed, and these can be useful in respiratory diseases. Examples are: apocynin, which by nebulization reduces H_2_O_2_ and the reduction of nitrites in the exhaled breath condensate of COPD patients [222,223]; setanaxib, which is an NOX1/4 inhibitor currently under development that has demonstrated excellent tolerability and reduction of various markers of chronic inflammation [224]. In the lung, other major enzymic antioxidants are superoxide dismutases (SOD), catalase and glutathione peroxidase (GPx). Recently, other enzymes have been found that play a crucial role in the lung antioxidant defense mechanisms: heme oxygenase-1 and small molecular weight redox proteins such as thioredoxins, peroxiredoxins and glutaredoxins [204].

The dismutation by SOD is of primary importance for each cell and is widely expressed in human lung/blood. The SOD enzyme helps convert superoxide to H_2_O_2_ and enzymes such as catalase and glutathione peroxidase (GPx) metabolized it [225]. SOD exists in three kinds of forms: (1) Copper zinc superoxide dismutase (Cu, Zn SOD) is mainly a cytosolic enzyme and abundant in most tissues. In particular, it is highly expressed in bronchial and alveolar epithelial cells and in mesenchymal cells, fibroblasts, and endothelial cells; (2) manganese superoxide dismutase (MnSOD) is mainly a mitochondrial enzyme and is expressed in alveolar type II epithelial cells and alveolar macrophages; and (3) extracellular superoxide dismutase (ECSOD) is primarily an extracellular enzyme that has been detected in plasma, airway epithelial cells (AEC) and alveolar macrophages (AM) [226]. Some studies show that SOD activity is reduced in the bronchial epithelium, in the cells in the bronchoalveolar fluid and in bronchial brushing of asthmatic patients compared with control subjects [217,227]. The newly characterized SOD mimetics appear to have beneficial effects for lung disorders caused by oxidants in animal models [217].

The CAT enzyme is an antioxidant found in many living tissues that use oxygen. It uses iron or manganese as a cofactor and catalyst and reduces hydrogen peroxide (H_2_O_2_) in water and molecular oxygen. Thus, the CAT enzyme completes the dismutation reaction affected enzymatically from SOD [228]. Studies indicate that metformin improves the levels of gene and enzymatic activities of CAT by protein kinase activated by AMP (AMPK), reducing OS [229,230,231]. The AMPK activator increased Nrf2 activation in an in vitro model of normal human bronchial epithelial cells (NHBE) stimulated with cigarette smoke extract, suggesting its potential protective action on lung inflammatory responses in severe COPD [232].

Glutathione peroxidase (GPxs) is an important enzyme, belonging to the oxidoreductase family, containing selenium that protects cells from damage caused by lipid peroxide and H_2_O_2_ [233]. In subjects with asthma and COPD, GPx activity is significantly reduced, and this is related both to the FEV1 values and to the body mass index of these subjects [234]. An example of antioxidant that exploits this mechanism is Ebselen [235], an organic selenium compound that acts as a GPx mimetic, effective in reducing inflammation and the release of inflammatory cytokines in the lungs of mice exposed to cigarette smoke. Although there are no data yet on its protective role in asthma or COPD, Ebselen could provide a new means of treating the life-threatening pulmonary and cardiovascular manifestations associated with cigarette smoking and COPD [236].

Antioxidant enzyme defense systems (including SOD, CAT, GPx, reduced glutathione and heme oxygenase 1) are directly regulated by Nrf2, which becomes a potential therapeutic target in lung diseases such as Idiopathic Pulmonary Fibrosis, asthma, COPD and acute respiratory distress syndrome [237,238]. Sulforaphane (a compound extracted from broccoli, cabbage and Brussels sprouts) was found to be an Nrf2 activator; experiments on human macrophages or mouse models suggest a preventive effect on COPD exacerbation [239]. Moreover, it modulates various signaling pathways associated with oncogenic EMT (epithelial-mesenchymal transition) [240]. Clinical studies have evidenced elevated Nrf2 expression in the lungs of patients with IPF, but these data need to be further supported by larger studies [241]. The intake of a dietary antioxidant such as vitamin C (ascorbic acid), vitamin E (α-tocopherol), resveratrol and flavonoids are suggested as antioxidant treatments in airway diseases [242,243]. Various epidemiological studies show that lung function (in terms of FEV1 and FVC) is improved by a high intake of antioxidants in the diet, resulting in a lower prevalence of chronic bronchitis and dyspnea [244]. Subjects with COPD (compared to healthy controls) who follow a diet with lower antioxidant content (low proteins, defective intake of iron, calcium, potassium, zinc, folate, vitamin B6, retinol) show a reduction in lung function with a higher risk of developing COPD [244,245]. The in vivo and in vitro models of inflammation induced by bleomycin, lipopolysaccharide and cigarette smoke demonstrate the antioxidant and anti-inflammatory properties of these compounds [246,247]. Additionally, some researchers show that high-intake dietary antioxidants protect against the progression of lung disease but this increases in mice exposed to cigarette smoke. These data suggest that indiscriminate use of dietary supplements could be a risk for the cure of lung diseases. For example, excessive use of vitamin E results in increased mortality in COPD patients [248]. OS activates kinases and redox-sensitive transcription factors, thereby modulating epigenetic changes in chromatin and causing changes in genetic transcription in COPD [249]. ROS (directly or indirectly) influence and potentiate inflammation in the lung via the activation of stress kinases such as JNK, MAPK, p38 and PI3K, or transcriptional factors as NF-κB, AP-1 and Nrf2 [241,250], strongly related with the pathogenesis of COPD, and asthma. Antioxidant agents are drug candidates useful to control these targets in various lung diseases [251].

The pharmacological treatment of COPD involves the administration of oxygen and oral or inhaled bronchodilators [252]. These last affect the redox balance, giving relief from the symptom, but are not resolutive. OS is one of the main causes of COPD by inducing PI3K activation and reducing activity of HDAC2 (histone deacetylase 2). HDAC2 interacts directly with proinflammatory transcription factors (TFs), such as NF-κB and AP-1, which alter the action of corticosteroids [253]. In fact, OS contributes to steroid resistance, as occurs for example in COPD [254]. However, corticosteroids are a powerful and universal drug treatment used in many lung diseases [213].

The action of corticosteroids is altered as interacting with glucocorticoid receptors (GR) form a complex that translocates into the nucleus and modulates the expression of NF-kB and AP-1 [250]. In normal conditions, HDAC2 deacetylates the GR, forming a protein–protein complex that represses the NF-kB pathway and attenuates the inflammation [255,256]. Hence, a reduction in HDAC2 expression does not deacetylatele the GR, which in turn cannot repress the NF-kB pathway [60]. Thus, OS is thought to be closely related to the development of steroid insensitivity in COPD [257]. One strategy used to bypass steroid resistance in airway diseases involves activating HDAC2 and reversing the post-translational oxidative modifications of HDAC2, which represents a possible therapeutic principle for the treatment of asthma and COPD [213,258] (Figure 3).

The excess of OS can be toxic to cells and tissues; it is true that cellular homeostasis and ROS synthesized by normal cells are crucial for maintaining health. Current knowledge of the increased activity of OS in the airway diseases suggests the relevant contribution of antioxidants in the treatment of asthma and COPD. The use of antioxidant also may help the resolution of the inflammation as coadjuvant of the conventional therapy in the mechanism of corticosteroid resistance often present in patients with severe asthma and COPD. However, the pharmacological manipulation of antioxidants and the development of truly effective new drugs target ROS regulatory mechanisms, which could provide a ray of relief in the challenging context of chronic inflammatory lung disease.

## 8. Conclusions

In this review, we give a general and extensive overview of OS, oxidative sources and antioxidant mechanisms concerning the chronic inflammatory diseases of the airways. There is much evidence showing that the mechanisms of OS are heavily involved in the pathogenesis of airway diseases. In fact, the lung is constantly exposed to external oxidative compounds (cigarette smoke, allergen, toxicants, etc.) present in contaminated air or released from inflammatory cells recruited and activated in the airways during inflammatory processes.

We tried to summarize the complexity and the variability of the mechanisms of OS in inflammatory diseases of the lung such as asthma and COPD. This complexity, associated with many oxidative compounds, prevents an adequate standardization of OS biomarkers useful for detecting the levels of activation of OS mechanisms in the lungs of asthmatic and COPD patients. This aspect makes it extremely difficult to assess and to define the general contribution and the role of OS metabolites in the development or progression of airway diseases leading to the origin of asthma and COPD. In fact, often the increased levels of OS biomarkers in the airway promote the progression of disease toward higher disease severity damaging the lung function and the response to the conventional drugs used in the treatment of chronic inflammatory diseases of the lung. However, often the treatment of inflammatory airway diseases with antioxidants as additional drugs is not sufficient to obtain the welfare of the patients, and it is necessary to deepen the research in the field of OS mechanisms.

Generally based on our research experiences, in this review we describe some knowledge on the mechanisms of OS that are especially associated with inflammation of the lung. Our aim is to stimulate the curiosity of readers to generate new scientific perspectives concerning the therapeutic action of antioxidants in the treatment of asthma and COPD, developing approaches that consider individual and environmental risk factors of patients that are useful in defining some concept of precision medicine.

## Figures and Tables

**Figure 1 antioxidants-11-02237-f001:**
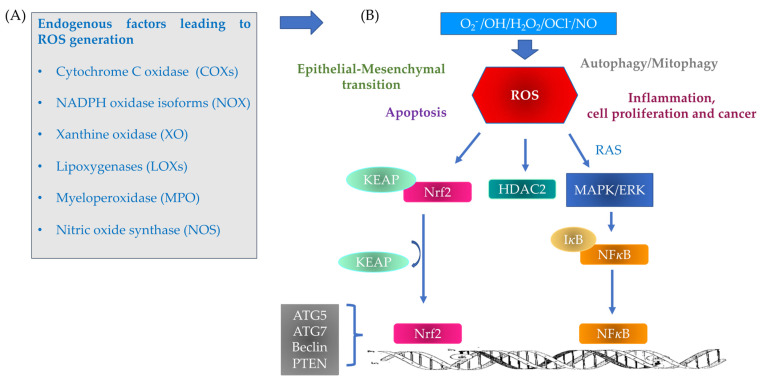
ROS led to activation of several signaling events including mitogen- activated protein kinase (MAPK) pathways, NF-E2-related factor (Nrf2)-mediated activation of nuclear factor-kB (NF-kB). In addition, ROS through Nrf2 influences proteins involved in autophagy/mitophagy (like, ATG5, ATG7, Beclin, and PTEN). Thus, ROS signaling events play a central role in regulation of proinflammatory events, proliferation, epithelial-mesenchymal transition and autophagy/mitophagy mechanisms. (**A**) Endogenous factors leading to reactive oxygen species (ROS) generation through: (I) highly reactive free radicals, including the superoxide anion (O_2_^_^), the hydroxyl radical (●OH) and (II) non-radical species such as hydrogen peroxide (H_2_O_2_), singlet oxygen (O_2_) [4,5], ozone (O_3_), hypochlorite anion (OCl^−^) and nitric oxide (NO). (**B**) Schematic depiction of multiple signaling pathways that generate ROS and the intracellular events activated by ROS accumulation.

**Figure 2 antioxidants-11-02237-f002:**
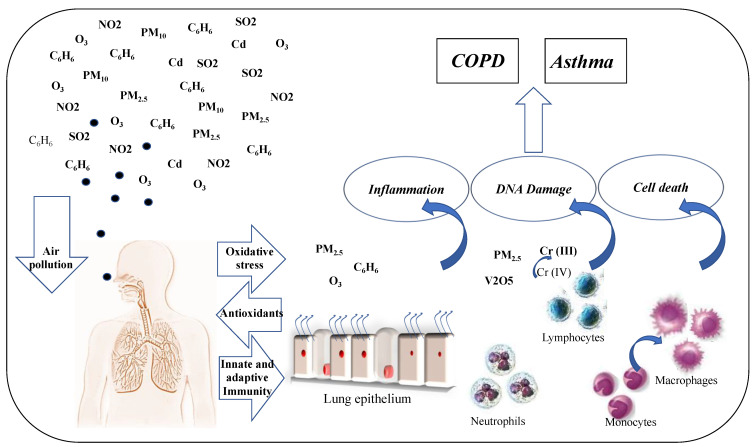
Effects of environmental pollution on OS damaging the activity of immune and structural cells in the airways of patients with chronic inflammatory diseases.

**Figure 3 antioxidants-11-02237-f003:**
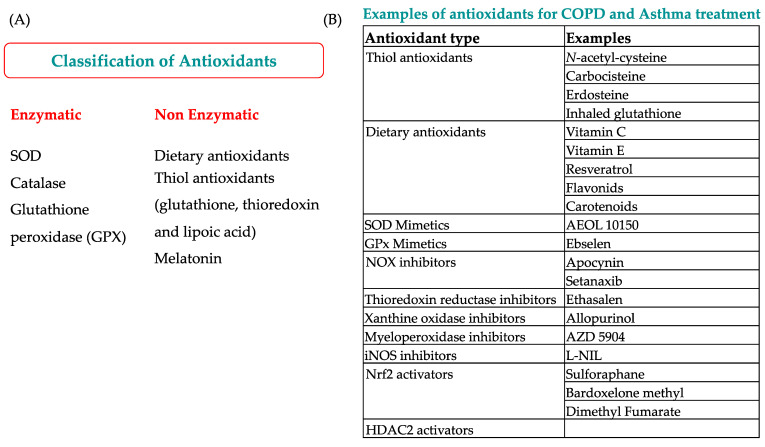
Antioxidants. (**A**) Classification of enzymatic and non-enzymatic antioxidants and (**B**) Examples of antioxidants for COPD and asthma treatment.

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
