# Peer review of "Overview of the Mechanisms of Oxidative Stress: Impact in Inflammation of the Airway Diseases"

_antioxidants, 2022, doi:10.3390/antiox11112237_

Round 1
Reviewer 1 Report
This review claiming to give an overview about the mechanisms of oxidative
stress and their implications in airway diseases contains unfortunately a lot of inaccuracies and flaws. Specifically, I have the following critical remarks:
1. Abstract + line 94 : the radicals are written as O2.- , OH. and NO. .
2. Line 113 and Fig. 1: Cytochrome c oxidase does not produce radicals at all, mitochondria produce superoxide at complex I (FMN-site) and complex III (Qi center of Q cycle).
3. Line 114: correct to NO synthase
4. Line 210: incorrect language.
5. Fig. 2 is missing, Fig. 3 appears twice. First Fig. 3 is perhaps Fig. 2, which is too primitive for a scientific journal.
6. Second Fig. 3B: What the authors mean with inhaled GSH? GSH is a threepeptide which cannot be evapurated.
7. Figs. 2 and 3 are not mentioned in the text at all.
8. The pharmacological perspectives paragraph needs to be restructured to follow a red line.
9. The conclusions are much too general and provide very little information, they need much more specification with respect to known facts and perspectives.
Reviewer 2 Report
The review on this topic is quite comprehensive. The authors basically talked about the OS in asthma, COPD and environmental pollution. These were more focused on the impact of OS on airway inflammation. How about airway remodeling and fibrosis? The authors are advised to add one more section to discuss OS in pulmonary fibrosis as fibrosis is closely related to chronic asthma, COPD and other types of fibrosis such as IPF.
Another issue that I would like to point out is regarding the HDAC2 section located on lines 706 - 713. The role of HDAC2 in executing corticosteroid's anti-inflammatory action is not clearly described and referenced. The authors may want to expand a bit more about this part. Also in Figure 3B, it should be HDAC2 activators instead of inhibitors.
A minor mistake for Figure "Effects of Environmental pollution and OS in the airways" labeling, it should be Figure 2 instead of Figure 3.
Round 2
Reviewer 1 Report
The authors substantially revised the previous version of the manuscript and improved the scientific soundness. However, still a careful check of chemical formulas (mainly of gases, cf. lines 556, 557, 575) and of spelling is required.
Author Response
Review report
Manuscript number: antioxidants-1945622
Title: Overview of the mechanisms of oxidative stress: impact in inflammation of the airway diseases.
Journal: Antioxidants
Reviewer 1
COMMENTS FROM EDITORS AND REVIEWERS General comment reviewer #1:
The authors substantially revised the previous version of the manuscript and improved the scientific soundness. However, still a careful check of chemical formulas (mainly of gases, cf. lines 556, 557, 575) and of spelling is required.
Response to the general comment: We thank the reviewer for the kind general comment about the content of the Manuscript. Accordingly with your comment, we revised the manuscript as you can see from the marked parts.
